# The Fewer the Merrier: Pruning Preferred Operators with Novelty

**Alexander Tuisov**[1] , **Michael Katz**[2]

[1]Technion, Haifa, Israel
[2]IBM Research, Yorktown Heights, NY, USA
alexandt@campus.technion.ac.il, michael.katz1@ibm.com

## Abstract

Heuristic search is among the best performing approaches to classical satisficing planning, with its performance heavily relying on informative and fast heuristics, as well as search-boosting and pruning techniques. While both heuristics and pruning techniques have gained much attention recently, search-boosting techniques in general, and preferred operators in particular have received less attention in the last decade. Our work aims at bringing the light back to preferred operators research, with the introduction of preferred operators pruning technique, based on the concept of novelty. Continuing the research on novelty with respect to an underlying heuristic, we define preferred operators for such novelty heuristics. For that, we extend the previously defined novelty concepts to operators, allowing us to reason about the novelty of the preferred operators. Our experimental evaluation shows the practical benefit of our suggested approach, compared to the currently used methods.

## 1 Introduction

Classical planning is among the most important areas of artificial intelligence. The performance of satisficing heuristic search based classical planners heavily relies on informative and fast heuristics, as well as search-boosting and pruning techniques. Recent advances in heuristics for classical planning [Keyder *et al.*, 2014; Domshlak *et al.*, 2015] allowed to go beyond delete relaxation and were responsible for the success of several satisficing planners such as *Mercury* [Katz and Hoffmann, 2014], *MERWIN* [Katz *et al.*, 2018], *Cerberus* [Katz, 2018], IBaCoP-2018 [Cenamor *et al.*, 2018], *OLCFF* [Fickert and Hoffmann, 2018], and *Saarplan* [Fickert *et al.*, 2018]. Search pruning techniques have also received some attention recently, mostly due to the development of *novelty*-based pruning techniques [Lipovetzky and Geffner, 2012; Lipovetzky and Geffner, 2014; Lipovetzky and Geffner, 2017b; Groß *et al.*, 2020] and novelty-based heuristics [Lipovetzky and Geffner, 2017a; Katz *et al.*, 2017; Groß *et al.*, 2020]. Search boosting techniques, on the other hand, mostly focused on introducing

randomness into the search [Valenzano *et al.*, 2014; Xie *et al.*, 2014], while the research on preferred operators [Hoffmann and Nebel, 2001; Richter and Helmert, 2009] was somewhat abandoned. To our knowledge, the most recent work introducing new preferred operators dates back to 2011, introducing preferred operators for an admissible heuristic, enabling the somewhat successful use of these admissible heuristics in a satisficing search [Bahumi *et al.*, 2011]. The recent partial delete relaxation heuristics, despite significantly extending the famous FF heuristic [Hoffmann and Nebel, 2001], when using preferred operators, use those of FF.

There is no doubt, however, in the practical usefulness of preferred operators. Arguably the most famous satisficing planners LAMA [Richter and Westphal, 2010] and FF [Hoffmann and Nebel, 2001] owe their success, at least in part, to the use of preferred operators. The empirical investigation of Valenzano *et al.* [2014] provides additional evidence of the importance of preferred operators. Thus, further research on preferred operators is of high potential value. To provide an anecdotal support for that claim, let's look at one of the most basic configurations with preferred operators, a greedy best-first search with two queues, all nodes, ordered by FF heuristic [Hoffmann and Nebel, 2001] and nodes obtained from preferred operators of FF. In our preliminary experiments, the configuration achieves a coverage of 1281 on the satisficing suite of the International Planning Competition (IPC) domains, while if a minor modification is made to the configuration, choosing each preferred operator with probability $0.25$, the coverage increases to 1387.6. Admittedly, when moving to more complex configurations, such as LAMA [Richter and Westphal, 2010], the effect of randomly pruning preferred operators is reduced significantly, and sometimes reduces the overall coverage. Choosing preferred operators with probability $0.25$ for FF, landmark count, or both results in coverage of 1581.4, 1603.0, or 1552.6, respectively, compared to 1594 for LAMA. Our conjecture in this work is, therefore, that a systematic method of pruning preferred operators can significantly improve planning systems performance.

In this work we present for the first time such a method, by introducing preferred operators for the novelty heuristic. We build upon the work of Katz *et al.* [2017], which adapts

the concept of novelty to heuristic search, by specifying the novelty of facts with respect to a heuristic. Here, we adapt the concept of novelty with respect to the underlying heuristic to operators, introducing the notion of novelty score for an operator. We exploit the new notion for systematically pruning preferred operators, introducing multiple definitions of preferred operators for the novelty heuristic. Our empirical evaluation shows a clear benefit of using novelty for pruning preferred operators of the underlying heuristic, compared to using all preferred operators from the underlying heuristic, as was done in previous work. Further, the empirical evaluation shows that pruning preferred operators with novelty is preferable to random pruning.

## 2 Preliminaries

In this work, we follow the notation of Bäckström and Nebel [1995]. A SAS$^+$ *planning task* is represented by a tuple $\langle \mathcal{V}, O, s_0, s_\star \rangle$, with $\mathcal{V}$ being a finite set of *state variables* and $O$ being a finite set of *operators*. Each state variable $v \in \mathcal{V}$ has a finite domain $dom(v)$ of values. A pair $\langle v, \vartheta \rangle$ with $v \in \mathcal{V}$ and $\vartheta \in dom(v)$ is called a *fact*. A (partial) assignment to $\mathcal{V}$ is called a *(partial) state*. Often it is convenient to view partial state $p$ as a set of facts with $\langle v, \vartheta \rangle \in p$ if and only if $p[v] = \vartheta$. A partial state $p$ is *consistent* with a state $s$ if $p \subseteq s$. We denote the set of all states of a planning task by $\mathcal{S}$. $s_0$ is the *initial state*, and the partial state $s_\star$ is the *goal*. Each *operator* $o$ is represented by a pair $\langle pre(o), eff(o) \rangle$ of partial states called *preconditions* and *effects*. An *operator cost* is a mapping $C : O \to \mathbb{R}^{0+}$. An operator $o$ is applicable in a state $s \in \mathcal{S}$ if and only if $pre(o)$ is consistent with the state $s$. Applying $o$ changes the value of $v$ to $eff(o)[v]$, if defined. The resulting state is denoted by $s[\![o]\!]$. An operator sequence $\pi = \langle o_1, \ldots, o_n \rangle$ is applicable in $s$ if there exist states $s_1, \cdots, s_{n+1}$ such that (i) $s_1 = s$, and (ii) for each $1 \leq i \leq n$, $o_i$ is applicable in $s_i$ and $s_{i+1} = s_i[\![o_i]\!]$. We denote the state $s_{n+1}$ by $s[\![\pi]\!]$. $\pi$ is a plan for the state $s$ iff $\pi$ is applicable in $s$ and $s_\star$ is consistent with $s[\![\pi]\!]$. The cost of a plan $\pi$, denoted by $C(\pi)$ is the summed cost of the actions in the plan. Classical planning deals with the problem of finding a plan for the initial state $s_0$.

A *heuristic function* is a mapping $h : \mathcal{S} \to \mathbb{R}^{0+}$, with $h(s)$ estimating the cost $C(\pi)$ of some plan $\pi$ for $s$. In addition to providing estimates for states, heuristics can identify a subset of applicable operators as *preferred*. The term *preferred operators* was coined by Helmert [2006], but was preceded by the term *helpful actions* [Hoffmann and Nebel, 2001], defined for the FF heuristic. For FF, helpful actions for a state $s$ are defined as the operators from the relaxed plan that are applicable in $s$. Similarly, preferred operators are defined for the additive [Bonet and Geffner, 2001] and the causal graph heuristics [Helmert, 2004], as well as for their generalization, the context-enhanced additive heuristic [Helmert and Geffner, 2008]. Preferred operators were also developed for the landmarks count heuristic [Richter *et al.*, 2008; Richter and Helmert, 2009; Richter and Westphal, 2010]. Landmarks are formulas that must be made true along any plan. Preferred operators for landmarks are applicable operators from a relaxed plan that achieves some

next achievable landmark. For implicit abstraction heuristics [Katz and Domshlak, 2010], preferred operators are those that start an abstract plan for at least one abstraction [Bahumi *et al.*, 2011].

Given a heuristic function $h$, the preferred operators of $h$ in state $s$ are denoted by $\mathcal{PO}_h(s) \subseteq O$.

A *search history* $\mathcal{H}$ is a set of pairs of operators and states that these operators lead to, starting with $\langle \emptyset, s_0 \rangle$. For each $\langle o, s \rangle \in \mathcal{H}$ such that $\langle o, s \rangle \neq \langle \emptyset, s_0 \rangle$, there exists another pair $\langle o', s' \rangle \in \mathcal{H}$ such that $s = s'[\![o]\!]$. Given an operator $o \in O$, the set of all states in the search history that $o$ leads to is denoted by $\mathcal{H}(o) := \{s \mid \langle o, s \rangle \in \mathcal{H}\}$. The set of all states in the search history is denoted by $\hat{\mathcal{H}} := \{s \mid \langle o, s \rangle \in \mathcal{H}, o \in O\}$, and the set of states in the search history that contain the fact $f$ is denoted by $\mathcal{H}(f) := \{s \in \hat{\mathcal{H}} \mid f \in s\}$.

For the concepts of *novelty*, we follow the notation of Katz *et al.* [2017], slightly adapting their definitions to the notion of search history defined above. We start with the definition of the *novelty score of a fact*.

**Definition 1** (heuristic novelty)**.** *Given a heuristic function* $h : \mathcal{S} \mapsto \mathbb{R}^{0+}$ *and a search history* $\mathcal{H}$, *the* **novelty score of a fact** $f$ *is defined as*

$$N(f, \mathcal{H}, h) = \begin{cases} \min_{s \in \mathcal{H}(f)} h(s), & \mathcal{H}(f) \neq \emptyset \\ \infty, & \text{otherwise.} \end{cases}$$

*Given a state* $s$, *the* **novelty score of a fact** $f$ **in state** $s$ *is defined as* $N(f, s, \mathcal{H}, h) = N(f, \mathcal{H}, h) - h(s)$ *if* $f \in s$.

To simplify the notation, we sometimes do not mention the search history $\mathcal{H}$ and the heuristic $h$ when these are clear from the context. A fact is novel in state $s$ if its novelty score in $s$ is strictly positive. A state is novel if it contains at least one novel fact. Katz *et al.* [2017] define a variety of novelty based heuristics, starting with the most basic one, separating novel states (that obtain the value 0) from the non-novel states (that obtain the value 1). Formally,

$$h_{BN}(s) = \begin{cases} 0, & \exists f \in s, N(f, s) > 0 \\ 1, & \text{otherwise.} \end{cases}$$

Going beyond this dichotomy, let $N^+(f, s)$ be 1 when $N(f, s) > 0$ and 0 otherwise. Similarly, let $N^-(f, s)$ be 1 when $N(f, s) < 0$ and 0 otherwise. Then, the second heuristic function also separates novel states, based on the number of novel facts. Formally,

$$h_{QN}(s) = |\mathcal{V}| - \sum_{f \in s} N^+(f, s).$$

Another heuristic function also separates non-novel states, based on the number of strictly non-novel facts.

$$h_{QB}(s) = \begin{cases} h_{QN}(s), & h_{QN}(s) < |\mathcal{V}| \\ |\mathcal{V}| + \sum_{f \in s} N^-(f, s), & \text{otherwise.} \end{cases}$$

While Katz *et al.* [2017] define additional heuristics, $h_{QB}$ was found to be best performing overall in their experiments.

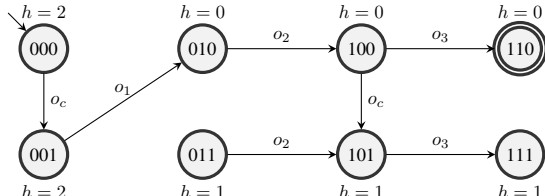

Figure 1: The state transition system of our running example.

# 3 Preferred Operators of Novelty Heuristics

We start by presenting a running example, a SAS$^+$ planning task with three binary variables $A, B, C$ and four operators $O = \{o_c, o_1, o_2, o_3\}$ as follows.

- $o_c = \langle \{B = 0, C = 0\}, \{C = 1\} \rangle$,
- $o_1 = \langle \{B = 0, C = 1\}, \{B = 1, C = 0\} \rangle$,
- $o_2 = \langle \{A = 0, B = 1\}, \{A = 1, B = 0\} \rangle$, and
- $o_3 = \langle \{A = 1, B = 0\}, \{B = 1\} \rangle$.

For the brevity of presentation, a triplet $abc$ denotes the state $A = a, B = b, C = c$. The initial state is therefore 000, and the goal state is 110. The full transition system, as well as the heuristic values of states are depicted in Figure 1. Finally, consider the following history $\mathcal{H} = \{\langle \emptyset, 000 \rangle, \langle o_c, 001 \rangle, \langle o_1, 010 \rangle, \langle o_2, 100 \rangle\}$. The current state is 100, and assume that $\mathcal{PO}_h(100) = \{o_c, o_3\}$.

Following the notation and the definitions presented in the previous section allows us to directly define *heuristic novelty* of operators, analogously to how a novelty of a fact is defined in Definition 1 [Katz *et al.*, 2017].

**Definition 2** (operator novelty score)**.** *Given a heuristic function* $h : \mathcal{S} \mapsto \mathbb{R}^{0+}$ *and a search history* $\mathcal{H}$, *the* **novelty score of an operator** $o$ *is defined as*

$$N(o, \mathcal{H}, h) = \begin{cases} \min_{s \in \mathcal{H}(o)} h(s), & \mathcal{H}(o) \neq \emptyset \\ \infty, & \text{otherwise.} \end{cases}$$

*Further, given a state* $s$, *the* **novelty score of an operator** $o$ **in state** $s$ *is defined as* $N(o, s, \mathcal{H}, h) = N(o, \mathcal{H}, h) - h(s)$.

In words, the novelty score of an operator in a state is the difference between the (best) heuristic value of a state previously reached by the operator during search and the heuristic value of the current state. Using our running example, $N(o_3, 100, \mathcal{H}, h) = N(o_3, \mathcal{H}, h) - h(100) = \infty - 0 = \infty$, while $N(o_c, 100, \mathcal{H}, h) = N(o_c, \mathcal{H}, h) - h(100) = 2 - 0 = 2$. Intuitively, larger (positive) novelty values correspond to situations where the operator, if previously applied, lead to states further away from the goal, according to the heuristic. Negative values mean that the operator was already applied during search, leading to states closer to goal than the current state, according to the heuristic. The heuristic, however, can be misleading, and therefore the boundary between considering an operator to be novel or not does not have to be at 0. A finer control of the threshold on novelty score for considering an operator to be novel may be beneficial. We say that an operator is *b-novel* in state $s$ if its novelty score in $s$ is greater than some predefined parameter

$b$: $N(o, s) > b$. Since the novelty scores can be negative, we allow $b$ to be negative as well. Setting $b = -\infty$ allows us to ignore the threshold when necessary.

We can now proceed with formally defining preferred operators for the novelty heuristic.

**Definition 3** ($b$-novel preferred operators)**.** *Given a heuristic function* $h$ *and a novelty score threshold* $b$, *the* $b$-novel *preferred operators of* $h$ *are defined as*

$$\mathcal{PO}_b(s, \mathcal{H}) = \{o \in \mathcal{PO}_h(s) \mid N(o, s, \mathcal{H}, h) > b\}$$

As per our running example, $\mathcal{PO}_2(100, \mathcal{H}) = \{o_3\}$, but $\mathcal{PO}_1(100, \mathcal{H}) = \{o_3, o_c\}$.

Definition 3 allows to select a subset of the preferred operators reported by the heuristic $h$ based on their novelty at that step of the search. However, it can be overly permissive, especially for small values of $b$. To overcome the issue, we suggest to select top elements (according to the novelty score) of the set $\mathcal{PO}_b(s, \mathcal{H})$. Assuming that the set $\mathcal{PO}_h(s)$ is given, we can select $k$ elements as follows.

**Definition 4** ($k$-top $b$-novel preferred operators)**.** *Given a heuristic function* $h$, *a natural number* $k$, *and a novelty score threshold* $b$, $\mathcal{PO}_b^k(s, \mathcal{H}) \subseteq \mathcal{PO}_b(s, \mathcal{H})$ *is the set of* $k$-top $b$-*novel preferred operators of* $h$ *if*

*(i) for all operators* $o \in \mathcal{PO}_b^k(s, \mathcal{H})$, *if there exists* $o' \in \mathcal{PO}_b(s, \mathcal{H})$ *such that* $N(o, s, \mathcal{H}, h) < N(o', s, \mathcal{H}, h)$, *then* $o' \in \mathcal{PO}_b^k(s, \mathcal{H})$, *and*

*(ii)* $|\mathcal{PO}_b^k(s, \mathcal{H})| \leq k$, *with* $|\mathcal{PO}_b^k(s, \mathcal{H})| < k$ *implying* $\mathcal{PO}_b^k(s, \mathcal{H}) = \mathcal{PO}_b(s, \mathcal{H})$.

On the other hand, Definition 3 may also be overly restrictive, since setting a finite threshold on novelty score may result in an empty subset of a non-empty set $\mathcal{PO}_h(s)$. To overcome the issue, we suggest to take the best operators in terms of novelty score, ignoring the threshold.

**Definition 5** (max-novel preferred operators)**.** *Given a heuristic function* $h$, *the max-novel preferred operators of* $h$ *are defined as*

$$\mathcal{PO}_{\max}(s, \mathcal{H}) = \arg\max_{o \in \mathcal{PO}_h(s)} \{N(o, s, \mathcal{H}, h)\}$$

As per our running example, $\mathcal{PO}_{\max}(100, \mathcal{H}) = \{o_3\}$. Note that the Definitions 3 - 5 are invariant under the novelty heuristic selected.

The use of preferred operators for search in classical planning is mainly for deriving an additional queue, consisting of a subset of successors, reached by these preferred operators. Search algorithms are then alternating between a complete queue with all successors and the preferred operators queue. The rationale behind the approach is that greedily following preferred operators may lead to the goal quicker, without the need to explore all successors. Our conjecture is that in many cases, further limiting the incomplete subset of successors may lead to the goal even quicker.

The rationale behind our approach of using the novelty score is that, for a particular preferred operator $o$, if a state reached by $o$ was not already explored during search or

| Domains | $h_{QB}^{FF}$ | $\mathcal{PO}_{-3}$ | $\mathcal{PO}_{-2}$ | $\mathcal{PO}_{-1}$ | $\mathcal{PO}_{0}$ | $\mathcal{PO}_{1}$ | $\mathcal{PO}_{2}$ | $\mathcal{PO}_{3}$ | $\mathcal{PO}_{max}$ |
|---|---|---|---|---|---|---|---|---|---|
| agricola18 (20) | **12** | 11 | **12** | 10 | 11 | **12** | **12** | **12** | **12** |
| airport (50) | **47** | **47** | **47** | **47** | 46 | **47** | **47** | **47** | 45 |
| barman14 (20) | 19 | **20** | **20** | **20** | **20** | **20** | **20** | **20** | **20** |
| childsnack14 (20) | 1 | 1 | 3 | 3 | 0 | **4** | 3 | 3 | 2 |
| data-network18 (20) | 14 | 14 | 14 | 16 | 16 | **17** | **17** | **17** | 16 |
| depot (22) | 19 | 21 | 21 | 21 | 21 | **22** | 21 | 21 | **22** |
| floortile11 (20) | **8** | **8** | 7 | 7 | 7 | 7 | 6 | 6 | 7 |
| floortile14 (20) | **3** | 2 | 2 | **3** | **3** | 2 | 2 | 2 | 2 |
| logistics98 (35) | 30 | 30 | 30 | 30 | 31 | 32 | 33 | **34** | **34** |
| nomystery11 (20) | **18** | **18** | 17 | 16 | 15 | 17 | 17 | 17 | 15 |
| openstacks11 (20) | **20** | **20** | **20** | **20** | **20** | **20** | 18 | 18 | **20** |
| openstacks14 (20) | **20** | **20** | **20** | **20** | **20** | 18 | 14 | 14 | **20** |
| openstacks (30) | **30** | **30** | **30** | **30** | **30** | **30** | **30** | 28 | **30** |
| organic-synthesis-split18 (20) | 12 | **14** | **14** | **14** | 13 | **14** | 13 | 13 | 13 |
| parking14 (20) | 6 | 7 | 6 | 6 | 7 | 8 | **9** | **9** | **9** |
| pathways (30) | 28 | 27 | 26 | 27 | 26 | 25 | 26 | 27 | **30** |
| pipesworld-notankage (50) | 43 | 43 | 44 | 44 | **47** | **47** | 45 | 45 | 44 |
| pipesworld-tankage (50) | 43 | 42 | 44 | **46** | 45 | 45 | **46** | **46** | 44 |
| satellite (36) | 28 | 28 | 28 | 26 | 28 | 28 | **29** | 28 | 28 |
| scanalyzer08 (30) | 29 | **30** | **30** | **30** | **30** | **30** | 28 | 29 | **30** |
| scanalyzer11 (20) | 19 | **20** | **20** | **20** | **20** | **20** | 18 | 19 | **20** |
| snake18 (20) | 6 | 6 | 7 | 6 | 8 | 7 | **9** | 7 | 7 |
| spider18 (20) | 16 | **19** | **19** | **19** | **19** | 18 | 18 | 17 | 17 |
| storage (30) | 28 | 27 | 28 | 27 | 28 | 28 | 27 | 28 | **30** |
| termes18 (20) | 14 | **16** | **16** | **16** | 14 | 15 | **16** | 15 | 15 |
| tetris14 (20) | 15 | 16 | 16 | 16 | 17 | **19** | 17 | 18 | 17 |
| thoughtful14 (20) | **19** | 18 | 17 | 18 | 17 | 16 | 17 | 17 | 18 |
| tidybot11 (20) | 17 | 17 | 17 | 17 | **18** | 17 | **18** | 17 | **18** |
| transport11 (20) | 15 | 16 | 18 | **19** | 17 | 18 | 18 | **19** | 16 |
| transport14 (20) | 11 | 10 | 8 | 11 | 11 | **14** | 9 | 10 | 10 |
| trucks (30) | 21 | **23** | **23** | **23** | 22 | 21 | 21 | 21 | **23** |
| visitall14 (20) | 18 | 19 | 18 | 18 | 19 | 17 | 18 | 18 | **20** |
| woodworking11 (20) | 19 | 19 | 19 | 19 | 19 | 19 | 19 | 19 | **20** |
| **Fully solved (883)** | 883 | 883 | 883 | 883 | 883 | 883 | 883 | 883 | 883 |
| **Sum other (100)** | 70 | 70 | 70 | 70 | 70 | 70 | 70 | 70 | 70 |
| **Sum (1816)** | 1601 | 1612 | 1614 | 1618 | 1618 | **1627** | 1614 | 1614 | **1627** |

Table 1: Coverage comparison: novelty heuristic, tie breaking by FF, preferred operators from FF ($h_{QB}^{FF}$), compared to the preferred operators pruned by novelty.

that state is further away from the goal (according to the heuristic function used) than the current state, then $o$ should be preferred over other preferred operators. When queues are ordered by states' heuristic values, as in the case of greedy best first search, the novelty score of the preferred operator $o$ in the current state $s$ and the heuristic novelty of the state $s' = s[\![o]\!]$ that results from applying $o$ to $s$ are somewhat independent. Thus, there are cases when $o$ is novel (that is, has a high novelty score), while $s'$ is not, and vice versa. In such cases, preferring less novel states that are reached by more novel operators may be beneficial. Going back to our running example, applying $o_3$ in 100 will result in a state 110, which, although non-novel, is a goal state (novelty heuristics are not necessary goal aware). Applying the less novel operator $o_c$ will transition the system to 101, which is a novel state, but further away from the goal. However, if all these operators are considered preferred, the search will explore the more novel resulting states before the less novel ones. Thus, to allow to greedily prefer more novel operators with less novel resulting state, we do not consider the less novel operators as preferred.

## 4 Experimental Evaluation

To empirically evaluate our approach, we implemented it on top of the Fast Downward planning system [Helmert, 2006]. The experiments were performed on Intel(R) Xeon(R) Gold 6248 CPU @2.50GHz machines, with the time and memory limit of 30min and 4GB, respectively [1]. The benchmark set consists of all STRIPS benchmarks from the satisficing tracks of International Planning Competitions (IPC) 1998-2018, a total of 1816 tasks in 64 domains. All tested configuration perform a greedy best-first search with delayed evaluation and multiple queues.

To empirically validate our conjecture that systematic pruning of preferred operators with novelty can improve performance, we take as a baseline the best performing variant of Katz *et al.* [2017], $h_{QB}^{FF}$. We enhance it with a second queue, defined by preferred operators of its underlying

---
[1] The code is at https://github.com/IBM/FD-Novelty-PO

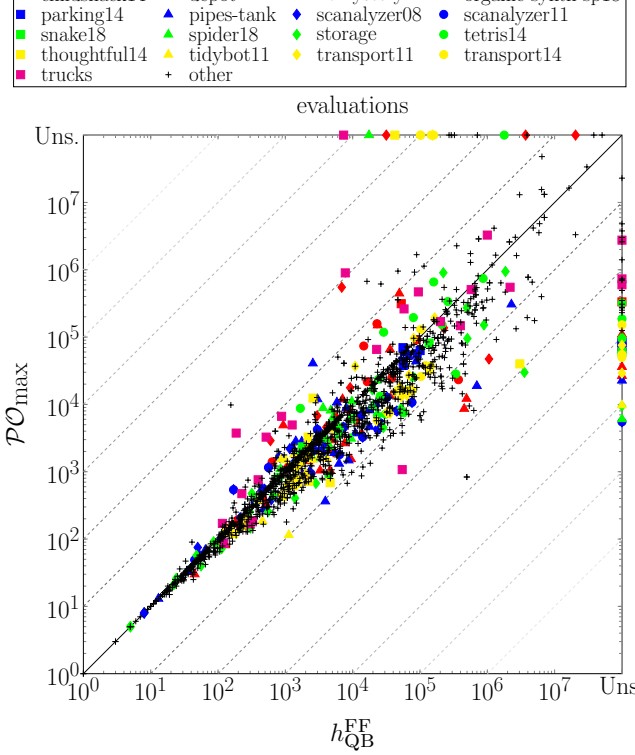

Figure 2: Per-task comparison of the number of evaluated states, greedy best-first search with two queues, heuristic values from $h_{QF}$ and preferred operators. The preferred operators $\mathcal{PO}_{max}$ are compared to the preferred operators from $h^{FF}$ ($h^{FF}_{QB}$).

| | $h^{FF}_{QB}$ | $\mathcal{PO}_{-3}$ | $\mathcal{PO}_{-2}$ | $\mathcal{PO}_{-1}$ | $\mathcal{PO}_0$ | $\mathcal{PO}_1$ | $\mathcal{PO}_2$ | $\mathcal{PO}_3$ | $\mathcal{PO}_{max}$ |
|---|---|---|---|---|---|---|---|---|---|
| All | **1601** | 1612 | 1614 | 1618 | 1618 | 1627 | 1614 | 1614 | 1627 |
| TL | 1600 | 1609 | 1611 | 1616 | 1615 | 1625 | 1614 | 1611 | |
| TM | 1599 | 1607 | 1612 | 1619 | 1619 | 1624 | 1617 | 1615 | |
| TS | 1598 | **1616** | **1616** | **1626** | 1623 | **1629** | **1627** | **1626** | |
| RL | 1599 | 1609 | 1613 | 1614 | 1620 | 1621 | 1615 | 1611 | 1627 |
| RM | 1599 | 1611 | 1614 | 1618 | 1619 | 1625 | 1618 | 1621 | 1626 |
| RS | **1601** | **1613** | **1621** | **1627** | **1632** | **1629** | **1623** | **1625** | **1629** |

Table 2: Overall coverage comparison: novelty heuristic, tie breaking by FF, preferred operators from FF ($h^{FF}_{QB}$), compared to the preferred operators pruned by novelty. The full set (All) is compared to selecting (i) top elements according to operator novelty score, large (TL), medium (TM), and small (TS) subsets, or (ii) randomly large (RL), medium (RM), and small (RS) subsets, values rounded to nearest integer.

heuristic, $h^{FF}$, with both queues being ordered by $h^{FF}_{QB}$, ties broken by $h^{FF}$.

Our first experiment compares the baseline to the configurations where the preferred operators of $h^{FF}$ are pruned either according to Definition 3, taking all preferred operators whose novelty score is above the threshold $b \in [-3, 3]$ (denoted by $\mathcal{PO}_b$), or according to Definition 5, taking all preferred operators with the maximal novelty score (denoted by $\mathcal{PO}_{max}$). Table 1 depicts a domain-wise comparison of the coverage for these configurations. There is a large portion of domains where all tasks are solved by all tested configurations. These domains are summarized in the row "Fully solved". Additionally, the row "Sum other" summarizes the domains where all configurations achieve the same coverage, but the domain is not fully solved. The remaining 33 domains are shown in the table. Finally, the last row summarizes the coverage for all domains.

While the overall coverage is improved for all tested configurations, on a per-domain level the baseline achieves the top performance on 9 out of the 33 domains. In one of these cases, on THOUGHTFUL14, all other configurations achieve strictly lower coverage. Focusing on $\mathcal{PO}_b$, while the best performing configuration overall is $\mathcal{PO}_1$, the best threshold varies for different domains. In fact, for some domains, there can be more than one best threshold. While we expected to observe a "parabolic" behavior, some domains

exhibit multiple peaks (e.g. AGRICOLA, CHILDSNACK, NOMYSTERY, TERMES18, and TRANSPORT11). In some other domains, larger thresholds are better (e.g. DATANETWORK18, LOGISTICS98, and PARKING14). In some, smaller thresholds work better (e.g. FLOORTILE11, OPENSTACKS, and TRUCKS). Thus, determining the best novelty score threshold for a planning task or even a domain can be beneficial. Switching our attention now to $\mathcal{PO}_{max}$, note that choosing only the most novel operators in each state performs as well overall as the best threshold. Comparing these two configurations, their strengths seem to be complementary, and a simple portfolio of these two approaches might significantly increase the overall coverage.

Going beyond pure coverage, Figure 2 shows a per-task comparison of the number of evaluated states between $\mathcal{PO}_{max}$ and the baseline configuration, grouped by domains. The domains where the median ratio of the number of evaluations was above 100 are emphasized, while the rest are aggregated under *other*. Out of the entire set of 1816 tasks, $\mathcal{PO}_{max}$ dominates in terms of evaluated states on 1064 tasks, while $h^{FF}_{QB}$ on 383 tasks. There are 19 tasks on the upper border, and 45 tasks on the right border (tasks solved by one approach but not the other). Although for most of the tasks the difference is within one order of magnitude, there are several cases where the difference is even more profound.

While the results of our first experiments are encouraging, our second experiment tests the conjecture behind Definition 4, that our configurations above can be overly permissive. Thus, we select top $k$ elements and test three bounds, $k \in \{10, 100, 1000\}$, denoting these by *small*, *medium*, and *large*, respectively[2]. Table 2, rows TL, TM, and TS show the overall coverage results, comparing to the results from our first experiment, depicted in the first row.

First, note that taking the top (according to the novelty score) from all preferred operators, without imposing any threshold (column $h^{FF}_{QB}$) does not significantly change the coverage. If the threshold is imposed, on the other hand,

---

[2]We have also experimented with relative bounds of 25%, 50%, and 75%, as well as minimal among the absolute and relative bounds, obtaining similar results.

the coverage increases consistently across our configurations when a small subset is chosen. When choosing a large or medium subsets, the coverage sometimes decreases, compared to choosing the entire set as in the first experiment. Focusing on the *small* subset and looking at the per domain results[3], probably the most notable change is in the OPENSTACKS domains, for $b = 2$ and $b = 3$, where all instances are now solved.

Our third experiment is intended to check whether the novelty score plays a significant role when selecting a subset of operators out of the novel operators $\mathcal{PO}_b$ or $\mathcal{PO}_{max}$. For that, we compared the selection of operators according to the novelty score to randomly choosing a subset, running each configuration 5 times and taking mean results. The overall coverage of these configurations, rounded to the nearest integer, is depicted in Table 2, rows RL, RM, and RS, where the first column depicts random pruning of operators. Focusing again on the best performer, the small subset, out of 7 thresholds, 3 achieve better mean performance, with the most notable change being in $\mathcal{PO}_0$, from 1623 for choosing a subset according to the top scores to 1632.2 for randomly choosing a subset. For $\mathcal{PO}_{max}$, since all operators in that set have the same novelty score, we can only compare to choosing the entire set (top row). The most notable gain is in PARKING14 (from 9 to 11.8) and in SATELLITE (from 28 to 32.4).

Finally, in order to evaluate the contribution of the new preferred operators to a state-of-the-art configuration, we compare to a state-of-the-art planner that uses the novelty heuristic, *Cerberus* [Katz, 2018]. Cerberus runs a greedy best-first search with two heuristics, novelty of the red-black planning heuristic and landmark count heuristic, with preferred operators from the red-black planning heuristic (that are essentially preferred operators from FF) and from the landmark count heuristic. For a cleaner comparison, we also compare to the variant that uses preferred operators from the red-black planning heuristic only, not using the preferred operators from the landmark count heuristic, denoted by $\mathcal{PO}_{h^{FF}}$. Our two suggested configurations replace the preferred operators $\mathcal{PO}_{h^{FF}}$ with $\mathcal{PO}_{max}$ and $\mathcal{PO}_1$, respectively. Table 3 depicts the per-domain coverage.

While there is now an even larger portion of domains where all tasks are solved by all three approaches (summarized in the row "Fully solved"), there is still a sufficient number of domains where performance in terms of coverage can still be improved (or reduced), 28 out of the total 64 domains. Out of these, Cerberus achieves top performance in 7 domains, with 3 of them not being matched by other approaches. Note that simply switching off preferred operators from the landmark count heuristic (column $\mathcal{PO}_{h^{FF}}$ in the table) improves the overall coverage by 13 tasks. Comparing $\mathcal{PO}_{h^{FF}}$ now to the two configurations that prune its list of preferred operators, note that both configurations improve the overall coverage, with $\mathcal{PO}_{max}$ increasing it by additional 32 tasks (overall, 45 tasks more than Cerberus). There are only 2 domains where $\mathcal{PO}_{h^{FF}}$ achieves a better coverage than $\mathcal{PO}_{max}$. Looking at the runner-up configuration $\mathcal{PO}_1$, note

---

[3]Detailed results can be found in the supplementary material.

| Domains | Cerberus | $\mathcal{PO}_{h^{FF}}$ | $\mathcal{PO}_{max}$ | $\mathcal{PO}_1$ |
|---|---|---|---|---|
| agricola18 (20) | 12 | 11 | **13** | 11 |
| airport (50) | 42 | 42 | **44** | 43 |
| childsnack14 (20) | **3** | 1 | **3** | 1 |
| data-network18 (20) | 14 | 13 | 15 | **16** |
| depot (22) | 21 | 20 | **22** | **22** |
| floortile11 (20) | **8** | 7 | 7 | 7 |
| floortile14 (20) | **3** | 2 | 2 | 2 |
| hiking14 (20) | 18 | 17 | **19** | 18 |
| nomystery11 (20) | **20** | 19 | **20** | 19 |
| openstacks11 (20) | 16 | **20** | **20** | **20** |
| openstacks14 (20) | 8 | **20** | **20** | **20** |
| organic-synthesis-split18 (20) | 12 | 14 | **15** | **15** |
| parking14 (20) | 13 | **20** | **20** | **20** |
| pathways (30) | 28 | 26 | **29** | 27 |
| pipesworld-notankage (50) | 43 | 43 | 45 | **47** |
| pipesworld-tankage (50) | 41 | 42 | **44** | 43 |
| rovers (40) | **40** | **40** | **40** | 39 |
| scanalyzer08 (30) | 29 | 29 | **30** | **30** |
| scanalyzer11 (20) | 18 | 19 | **20** | **20** |
| snake18 (20) | 9 | **10** | 8 | **10** |
| sokoban11 (20) | 17 | **18** | **18** | 17 |
| spider18 (20) | 18 | 16 | **19** | 18 |
| storage (30) | 26 | 26 | **29** | **29** |
| termes18 (20) | 12 | **13** | **13** | **13** |
| tetris14 (20) | **19** | **19** | 17 | **19** |
| thoughtful14 (20) | 16 | 16 | **20** | 18 |
| tidybot11 (20) | 17 | **19** | **19** | 18 |
| trucks (30) | **24** | 18 | 21 | 21 |
| **Fully solved (1024)** | 1024 | 1024 | 1024 | 1024 |
| **Sum other (80)** | 50 | 50 | 50 | 50 |
| **Sum (1816)** | 1621 | 1634 | **1666** | 1657 |

Table 3: Coverage comparison to the state of the art.

that it was able to obtain a better coverage than $\mathcal{PO}_{max}$ in 4 domains. Overall, the experiments clearly show the benefit of systematically pruning preferred operators, significantly improving the performance of even a state-of-the-art planning system.

## 5 Discussion and Future Work

We have shown in this work how to define preferred operators for the novelty heuristic, extending the notion of novelty with respect to an underlying heuristic to operators. Our experimental evaluation shows that the approach works well in practice, increasing the coverage in many domains, sometimes significantly. The notion of operator novelty is somewhat orthogonal to the notion of novel states. Non-novel operators can lead to novel states and novel operators can lead to non-novel states. Not considering non-novel operators as preferred allows us to better focus the greedy exploration and obtain better results. Our experiments also show that the best novelty score threshold can vary from one domain to another.

It is worth mentioning that our approach does not require to know operators preconditions or effects, only whether an operator is applicable in the state. Therefore, it can

potentially be applied to formalisms where no action model is available, such as black-box planning [Jinnai and Fukunaga, 2017; Lipovetzky *et al.*, 2015]. Additionally, we intend to explore ways of obtaining a novelty score threshold on a per-domain or per-instance basis. Further, additional possible definitions of preferred operators for the novelty heuristic can be obtained. Finally, we would like to further investigate the reason behind the improved performance, attempting to extend our understanding of the novelty based approaches in heuristic search for classical planning.

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
