# OpenReview forum: "The Fewer the Merrier: Pruning Preferred Operators with Novelty"
_icaps-conference.org/ICAPS/2021/Workshop/HSDIP — HSDIP 2021_

### Official Review · AnonReviewer1 · 2021-05-25

**Confidence:** 4
**Overall Score:** Accept

**Review:**

The paper introduces a method to prune preferred operators based on a novelty criterion similar to the heuristic novelty introduced by Katz et al. (2017).
The authors introduce several definitions of novel operators and evaluate their effectiveness on the IPC benchmarks.
Compared to a baseline using all preferred operators, using only the novel ones shows consistent improvements across most domains.

In my opinion, this is an interesting paper and the topic is a perfect fit for HSDIP.
The paper is well-written, and the contributions are well-motivated in theory and extensively evaluated in practice.
I have a few comments and questions below.

1. In the introduction, you write:
  > "The aforementioned recent partial delete relaxation heuristics, [...] when using preferred operators, use those of FF."

  I am not sure about red-black heuristics, but at least for conjunction-based methods, this statement seems incorrect.
  The preferred operators are extracted following the same methodology as that of FF (taking all applicable operators from the generated plan), but since the partially relaxed plan potentially differs from the fully relaxed plan of FF, so do the extracted preferred operators.

2. In the evaluation for $k$-top operator novelty you use the values $k = \{10, 100, 1000\}$. These values seem very large to me, as I would not expect any state to have that many successors through preferred operators (in most standard planning domains at least). Did you have some statistics on the average number of preferred operators across these domains to come up with these bounds? Have you tried smaller numbers for $k$ as well (e.g., in the range of 1 to 10)?

Minor comments:
- I would suggest using $\bot$ instead of $\emptyset$ for the initial state in the search history notatation (since $\emptyset$ is not an operator).
- The verbal description of the operator novelty (the sentence starting with "In words, ..." following Definition 2) suggests that it considers states in which this operator was applied, but I think it should be states that were reached through this operator.

---

> ### Author Response · Authors · 2021-06-02
> **"The aforementioned recent partial delete relaxation heuristics, [...] when using preferred operators, use those of FF."**
>
> While we agree that different implementations may well result in different sets of preferred operators (even for the FF heuristic itself), that was not the intention of the cited sentence. The sentence claims (rightfully), that on a conceptual level, both the red-black planning heuristic and the conjunction-based methods are still using conceptually the same reasoning for selecting preferred operators. While for red-black heuristic these preferred operators are literally taken from the underlying FF implementation, for the conjunction-based methods it might be different on the implementation level, but on the conceptual level these are still the preferred operators of the FF heuristic (possibly for a compiled task), as you say yourself.

---

> > ### Comment · AnonReviewer1 · 2021-06-04
> > **Preferred Operators of Partial Delete Relaxation Heuristics**
> >
> > To clarify, we're talking about:
> > (a) the methodology of deriving the preferred operators,
> > (b) the resulting set of preferred operators.
> >
> > We clearly agree that (a) is the same for partial delete relaxation heuristics and the FF heuristic. My point is that the current wording in the paper instead implies that (b) is the the same (conceptually, ignoring differences in implementation), which I don't agree with. Preferred operators that are selected from a partially relaxed plan will sometimes contain operators that would not be preferred by *any* (reasonable) FF implementation, i.e., it will sometimes contain an operator $o$ that is only included in the partially relaxed plan because it is used to achieve certain things that are treated with non-relaxed semantics, while fully relaxed plans would never include $o$.

---

> > > ### Author Response · Authors · 2021-06-04
> > > **We understand your point**
> > >
> > > We can clarify this point in the revised version.

---

> ### Author Response · Authors · 2021-06-02
> **Absolute bounds (please also see a reply to the other review)**
>
> Unfortunately, no, we do not have any statistics on the average number of preferred operators.
> Indeed, it would be interesting to explore smaller absolute bounds, in a more systematic way, in a future research.

---

### Official Review · AnonReviewer2 · 2021-05-27
**The paper generalizes the concept of novelty to operators and gives a thorough experimental analysis**

**Confidence:** 4
**Overall Score:** Accept

**Review:**

The paper introduces preferred operator pruning based on novelty for satisficing classical planning by generalizing the notion of novelty to operators, giving an operator the minimum novelty score of any state that it generated in the search history.
This does an additional pruning of the set of preferred operators obtained from an arbitrary heuristic with which the approach can be combined.

The authors do a thorough evaluation of their approach, showing that the new novelty-based preferred operator pruning is indeed effective and can increase the planner performance. It also shows, though, that it needs some further investigation. The random selection of a subset of the novel operators often outperforms the selection of the top-k novel operators.

The paper is well-written and easy-to-follow. It is technically sound and the topic fits HSDIP.

- Would it make sense to define operator novelty based on the effect facts instead of using the novelty of the generated states?

- The values chosen for b and k are not well motivated. In how many instances is there a difference at all when comparing k=100 and k=1000? Does moving to larger (absolute) values of b change the results? Does it make sense to chose b based on the number of variables?

- Below Def2: ".. of a state where the operator was previously applied during search"
Shouldn't it be the states that were *generated by* the operator?


Minor:
- Above Def.1: "search history that o leadS to"
- Def4: "..is the SET OF k-top b-novel.."
- Evaluation: "number of evaluations was above 100 are emphasizeD"
- "While there is now AN even larger portion.."

---

> ### Author Response · Authors · 2021-06-02
> **Define operator novelty based on the effect facts**
>
> We assume that you mean a new definition, something like operator novelty is equivalent to the novelty of the conjunction of the effect facts, and not modifying our definition to account for effect facts only instead of the entire resulting state (since it is not clear at all how that could be done).
>
> Such a definition would create a correlation between novel operators and novel outcome states, and therefore should not have an additional pruning power on top of the pruning/preference done on states.

---

> ### Author Response · Authors · 2021-06-02
> **The values chosen for b and k are not well motivated**
>
> While we have not checked in how many instance there is an actual difference between the k=100 and k=1000 configurations, there is a difference (as can be seen in the supplementary material).
>
> Indeed, our exploration of absolute and relative bounds was far from exhaustive and can (and should) be a topic for future work.
> It might make sense to learn a domain-dependent, instance dependent, or even state-dependent threshold. As a result, relevant properties might include domain/instance/state dependent features, including but not limited to the number of state variables.

---

> ### Author Response · Authors · 2021-06-02
> **Below Def2**
>
> Thank you for catching that. Indeed, there was a mistake in the text, the new text now reads
> "... the novelty score of an operator in a state is the difference between the (best) heuristic value of a state previously reached by the operator during search and the heuristic value of the current state."

---

### Decision · Program_Chairs · 2021-06-10

**Decision:**

Accept

**Comment:**

The reviewers agreed that the paper should be accepted and mentioned that there are certain points that should be clarified. We ask the authors to do their best to address these comments in the final version.